# Predictors of Basic Activity in Daily Living and Length of Hospitalization in Patients with COVID-19

**DOI:** 10.3390/healthcare10081589

**Published:** 2022-08-22

**Authors:** Ting-Jie I, Yu-Lin Tsai, Yuan-Yang Cheng

**Affiliations:** 1Department of Medical Education, Taichung Veterans General Hospital, Taichung 407219, Taiwan; 2Department of Physical Medicine and Rehabilitation, Taichung Veterans General Hospital, Taichung 407219, Taiwan; 3Department of Post-Baccalaureate Medicine, College of Medicine, National Chung Hsing University, Taichung 407224, Taiwan

**Keywords:** activity of daily living, COVID-19, mobility limitation, muscle strength

## Abstract

Background: Patients recovered from COVID-19 often suffer from the sequelae of the disease, which can hinder the patients’ activity in daily living. Early recognition of the patients at risk of prolonged hospitalization and impaired physical functioning is crucial for early intervention. We aim to identify the predictors of prolonged hospitalization and impaired activity in daily living in this study. Methods: COVID-19 patients hospitalized in a medical center were divided into two groups according to the Barthel index three months after discharge and the median length of hospital stay, respectively. Chi-square test and Mann–Whitney U test were performed to check the differences between the two groups in patient characteristics as well as hematology tests at the emergency department, the intensive care unit mobility scale (ICUMS), and the medical research council sum score (MRCSS). Logistic regression and the receiver operating characteristic curve analysis were further performed for the factors with significant differences between the two groups. Results: Both ICUMS and MRCSS showed significant differences between the groups. The ICUMS had an odds ratio of 0.61 and the MRCSS of 0.93 in predicting a Barthel index score less than 100 three months after discharge. The MRCSS had an odds ratio of 0.82 in predicting a prolonged length of hospital stay. Conclusion: Both ICUMS and MRCSS upon admission are predictive of a Barthel index score of less than 100 three months after discharge. On the other hand, only MRCSS has predictive value of a prolonged hospitalization.

## 1. Introduction

Ever since its initial outbreak, COVID-19 has been rapidly spreading throughout the world. As of 25 January 2022, there have been more than 350 million confirmed cases worldwide, causing more than five million deaths [1]. The calculated mortality rate from the virus is approximately 1.6%. Since most COVID-19 patients survive the illness, the sequelae of the disease have become an important issue. Previous studies have indicated that survivors of COVID-19 experience different levels of physical function impairment, pulmonary function damage, and even mood disorders [2,3,4]. These sequelae may cause certain disturbances to a patient’s daily life even after being discharged from hospital [3,4,5]. A previous study has indicated that early mobilization is beneficial for COVID-19 patients towards improving their sequelae in the aspects of physical function, cognitive function, and respiratory condition [6]. Since the sequelae are preventable, prediction and early recognition of the patients who may need timely intervention are necessary.

There have been studies that have attempted to find the predictive factors for the clinical outcomes of COVID-19 patients [7,8,9,10,11]. However, studies using physical performance as predictive factors were scarce in the past. Additionally, a patient’s functional prognosis after discharge is rarely assessed even though it may be hindered by the sequelae of COVID-19. van Gassel et al. observed that COVID-19 patients with better muscle strength and pulmonary function experience better physical functioning three months after discharge [12]. The outcome measurement used to assess muscle strength in this study was Medical Research Council Sum Score (MRCSS), which is the sum score of both proximal and distal muscle strength of four limbs. This score can provide a comprehensive strength evaluation of extremities and could be one of the possible predictors of physical function in the future. On the other hand, the study done by Pinto et al. concluded that physical activity, which was evaluated by Baecke Questionnaire of Habitual Physical Activity (BQHPA), has no independent association with clinical outcomes among hospitalized COVID-19 patients [13]. In contrast to BQHPA, which evaluates individuals’ habitual physical activities over the previous 12 months, Intensive Care Unit Mobility Scale (ICUMS) reflects the physical mobility at the timing of hospital admission and could be another predictive factor of functional outcomes. Because of the rarity of literature in the past, the ability of physical performance in predicting functional outcomes in COVID-19 patients deserves more studies for clarification.

The length of hospital stay is another determinant surrounding return to normal activities in one’s daily life. A previous study showed that a longer hospitalization period is associated with more functional decline [14]. Thus, identifying risk factors of a prolonged hospitalization in patients with COVID-19 is also a crucial issue. There are numerous studies that have investigated the risk factors regarding a prolonged hospitalization period [15,16,17]. Wu et al. [17] and Liu et al. [15] discovered that high fever, bilateral pneumonia seen on a chest computed tomography, diabetes, and lymphopenia can all predict longer hospitalization. However, physical performance has rarely been used to predict the length of hospitalization to date.

Physical performance can be evaluated through simple and easy clinical measurement tools by healthcare workers with no additional cost. However, according to our literature review as above, there were only few studies regarding the predictive ability of physical performance on the functional prognosis and length of hospitalization until now. Therefore, the purpose of this study is to find out whether a COVID-19 patient’s physical performance along with other laboratory data upon admission are associated with both functional prognosis and hospitalization stay duration. Based on a past study [18], which showed impaired baseline activities of daily living being associated with functional decline at discharge, we hypothesize that COVID-19 patients with a poor physical performance upon admission shall have longer hospital stays and worse functional outcomes.

## 2. Materials and Methods

### 2.1. Study Population

This single-centered, retrospective cohort study was approved by the Institutional Review Board of Taichung Veterans General Hospital. Consents from the patients were exempted due to the approval (CE21317A#1). The protocol was prespecified in advance and registered in the ClinicalTrials.gov (Identifier: NCT05195736). Hospitalized patients based on the initial diagnosis of COVID-19 between 1 January 2020 and 15 June 2021 were included in our study. These patients all registered a positive COVID-19 polymerase chain reaction test, pneumonia appearance on a chest X-ray, and prominent clinical symptoms related to COVID-19, thus fulfilling the criteria for hospitalization. Patients were excluded if they were less than 18 years old or had any medical history that impeded performing basic activities during daily life, such as fractures, stroke, or other neurodegenerative diseases. The Barthel index was used to check the basic activities of daily life of these patients [19]. Those with a baseline Barthel index score less than 100 before the onset of COVID-19 were excluded from our study.

### 2.2. Study Design

Patient characteristics, including age and sex, were collected. Data from the initial hematology tests taken at the emergency department, including white blood cell count (WBC), hemoglobin (Hb), total bilirubin, lactate dehydrogenase (LDH), D-dimer, blood urea nitrogen (BUN), creatinine (Cr), sodium (Na), potassium (K), calcium (Ca), phosphorus (P), magnesium (Mg), erythrocyte sedimentation rate (ESR), C-reactive protein (CRP), prolactin, alkaline phosphatase (ALKP), activated partial thromboplastin time (APTT), prothrombin time (PT), lactate, albumin, creatine kinase (CK), and creatine kinase-MB (CKMB), were all recorded. In addition, while focusing on the patients’ physical performance, the ICUMS and the MRCSS were both extracted by a research assistant via medical chart. The ICUMS was first developed by Hodgson et al. to evaluate the highest level of mobility in adult ICU (intensive care unit) patients [20]. The scale is ranked from zero to ten, where zero means no active movement at all by the patient, and ten means the patient can walk independently without any aid. The scoring criteria of one to nine are based on ability to performing the following works: (1) bed activities; (2) passive moving to a chair; (3) sitting over edge of bed; (4) standing; (5) active transferring to a chair; (6) marching on spot; (7) walking with assistance of two or more people; (8) walking with assistance of one person; and (9) independent walking with a gait aid [20]. Tipping et al. observed the predictive value of the ICUMS for ICU discharge survival and discharge destination [21]. The MRCSS, on the other hand, is composed of the muscle strength of moving the bilateral shoulders, elbows, wrists, hips, knees, and ankles, with each muscle being scored from zero to five, resulting a total score from zero to sixty [22]. It was initially used to assess muscle weakness in patients with Guillain–Barre syndrome [22]. Hermans et al. also found its reliability and reproducibility in evaluating muscle weakness in ICU patients [23]. It was also used to predict discharge destination in the study done by Perme et al. [24].

Two targets were selected as the outcomes of our study. First, the length of hospital stay was assessed based on the admission and discharge date logged in the medical records. Second, the Barthel index scores three months after discharge of the patients were assessed via phone calls. Patients were divided into two groups according to their outcomes, those with a hospitalization period longer than median or shorter than median, and those with a Barthel index score of 100 or less than 100 three months after discharge.

### 2.3. Statistical Analysis

We used the Statistical Package for the Social Science (SPSS) 16.0 (Chicago, IL, USA) for statistical analysis in our study. The chi-square and Mann–Whitney U tests were used to check the statistical difference between the two groups in the categorical data of sex, with continuous data including WBC, Hb, total bilirubin, LDH, D-dimer, BUN, creatinine, Na, K, Ca, P, Mg, ESR, CRP, prolactin, ALKP, APTT, PT, lactate, albumin, CK, CKMB, ICUMS, and MRCSS. Next, for those data showing significant differences between the two groups, logistic regression analysis was performed to identify the most important parameters in determining the length of hospitalization as well as the Barthel index three months after discharge. Furthermore, receiver operating characteristic (ROC) curve analysis was done to determine the area under the curve (AUC) and the cutoff points of these parameters. A *p*-value less than 0.05 was considered significant in our study.

## 3. Results

From 1 January 2020 to 15 June 2021, a total of 47 patients were enrolled in our study. Ten patients were excluded due to impaired baseline activities in their daily living, two were excluded because of age < 18, and four were lost to three-month follow-up after discharge. In the end, 31 patients were eligible for data analysis. The flowchart for the enrollment of our study subjects is shown in Figure 1. Among them, 16 were male, and 15 were female. The average age of the patients was 51.03 years old, ranging from 18 to 84 years old. The patients’ average Barthel index three months after discharge was 92.9 ± 19.74, with the median length of hospital stay being 20 days, with a range of 8 to 56 days. The characteristics and the initial blood profiles of our study subjects are summarized in Table 1. The mean and median values were both provided in order to present the data distribution of the enrolled patients in the study.

We used chi-square and Mann–Whitney U tests to check the differences between the groups divided by the Barthel index three months after discharge and the median of hospitalization period. Amongst the 31 enrolled patients, 6 did not reach a Barthel index score of 100 three months after discharge. Table 2 compares the age, sex, ICUMS, MRCSS, and blood profiles obtained at the emergency department prior to hospitalization between patients with a Barthel index score of 100 and those without. The age of patients was younger in the group with a Barthel index score of 100 (47.96 ± 18.45 versus 63.83 ± 17.0), but statistical significance was not achieved (*p* = 0.065). Additionally, both the ICUMS and the MRCSS were significantly higher in this group (ICUMS: 8.28 ± 2.79 versus 2.83 ± 3.13, *p* = 0.001; MRCSS: 50.96 ± 13.68 versus 29.33 ± 16.91, *p* = 0.015). The sex and blood profiles were not significantly different between the two groups. Table 3 compares these factors between the groups based on the length of hospital stay. The average age was younger in the group with a shorter hospitalization period (44.13 ± 20.69 versus 57.5 ± 15.17, *p* = 0.044). The ICUMS and the MRCSS were both significantly higher in patients with a shorter length of hospital stay (ICUMS: 9.53 ± 0.74 versus 5.06 ± 3.8, *p* = 0.001; MRCSS: 57.33 ± 4.7 versus 36.88 ± 17.57, *p* < 0.001). Sex and all the hematology test results failed to show significant differences between the two groups. The results of differences between the two outcomes were similar. Both ICUMS and MRCSS were significantly different between those with longer and shorter hospitalization and between those with full Barthel index scores three months after discharge and those without. While age was significantly different only between those with longer and shorter hospitalization, all the differences of sex and the blood profiles were not significant in both outcomes in our study.

Age, ICUMS, and MRCSS were then recruited for logistic regression. In univariate analysis, as shown in Table 4, after adjusting for age, we found that higher ICUMS and higher MRCSS were both negative predictive factors for a Barthel index score less than 100 at three months after discharge (ICUMS, OR = 0.61, *p* = 0.012; MRCSS, OR = 0.93, *p* = 0.024). Alternatively, as shown in Table 5, we found that MRCSS was a negative predictive factor (OR = 0.82, *p* = 0.02) for longer hospitalization. No significant predictive factors for either Barthel index scores less than 100 at three months after discharge or longer hospitalization could be identified in further multivariable regression analysis.

The ROC curve analysis was performed on ICUMS and MRCSS for the Barthel index scores three months after discharge and for the days of hospitalization. For identifying patients with a Barthel index score of less than 100 at three months after discharge, the ICUMS presented an AUC of 0.9, while the MRCSS presented an AUC of 0.803 (Figure 2). As for the length of hospital stay, the ICUMS presented an AUC of 0.838, and the MRCSS presented an AUC of 0.875 in identifying patients with hospitalization longer than 19 days (Figure 3). The cutoff points in predicting both outcomes are shown in Table 6. For ICUMS, the cutoff points for both Barthel index scores three months after discharge and the days of hospitalization was 6, which means the COVID-19 patients might be hospitalized for more than 19 days or have a Barthel index < 100 three months after discharge if their initial ICUMS score was less than six. On the other hand, for MRCSS, it was 40 and 50, respectively. It indicates that patients might not be totally independent three months after discharge if their initial MRCSS scored less than 40 and might be hospitalized for more than 19 days if their initial MRCSS scored less than 50. The results of the predictive ability and cutoff points are summarized in Figure 4.

## 4. Discussion

In this retrospective cohort study, we found significant differences in ICUMS and MRCSS between patients with impaired activities of daily living and those without. Patients with better ICUMS and MRCSS upon admission were less prone to develop functional limitation in our regression model after adjusting for age. Additionally, patients with better MRCSS upon admission tended to have shorter hospitalization. Finally, older patients in our study tended to have longer hospitalization, but age did not play a significant role in predicting activities three months after discharge and the length of hospitalization.

To the best of our knowledge, this is the first article focusing on the value of mobility and muscle strength in predicting COVID-19 patients’ functional outcomes. One of the most important functional outcomes, the basic activity of daily living, was assessed by the Barthel index in our study. Some items of the Barthel index were important for hospitalized patients, such as feeding, transfer, toileting, bowel management, and bladder control. Other items were more essential for discharged patients without caregivers, including grooming, dressing, mobility, and stairs climbing. The Barthel index has been used as an important indicator of hospital outcome [25] and could be used as a predictor of mortality [26]. On the other hand, there have been studies on patients with COVID-19 that aimed to identify predictive factors of functional prognosis [18,27,28]. Similar to our study, Hosoda et al. also discovered that baseline activities of daily living impairment was associated with a greater risk of functional decline at discharge [18]. Other risk factors were also identified, such as an age older than 60, more severe diseases, and more symptoms upon infection, all of which were revealed as risk factors for moderate dependence in activities of daily living as measured by the Barthel index at discharge [28]. Another study performed by Frontera et al. identified neurologic complications during hospitalization as being an independent predictor of limited activity in daily living for COVID-19 patients 6 months after hospitalization [27]. The mobility status and muscle strength of the four limbs are closely related to activities performed during daily living, and therefore, the results of our study are supported by the past literature [18]. In addition, our study extrapolated predictive ability to three months after discharge from hospital. The mechanism behind this may be related to the detrimental involvement of multiple organs in addition to the respiratory system that is caused by severe COVID-19. A more declined physical performance in the initial stage of COVID-19 signified a more advanced disease severity, which could therefore bring a subsequent longer length of hospitalization and more physical sequelae three months after discharge.

There have been numerous studies that have tried to find the predictors of the length of hospitalization in COVID-19 patients [15,16,17]. However, a patient’s mobility status and muscle strength are rarely considered. Our results are supported by the study of Gil et al., which revealed that both muscle strength and muscle mass are predictors of the length of hospital stay in patients with moderate-to-severe COVID-19 [29]. Their study used handgrip strength as a muscle strength measurement and the vastus lateralis cross-sectional area as a muscle mass measurement. In their study, patients at the highest one-third handgrip strength level experienced significantly shorter hospitalization stays. Alternatively, patients at the lowest one-third of the vastus lateralis cross-sectional area had significantly longer hospital stays [29]. Our study shares similarities with this previous study although muscle strength in our study was measured by MRCSS. Nevertheless, MRCSS is more sensitive to changes in certain clinical courses, such as Guillain–Barre syndrome [22], and can better evaluate global muscle condition than handgrip strength alone [30]. Moreover, the study done by Lee et al. revealed that, compared with handgrip strength, MRCSS served as a better independent predictor of mortality, length of surgical ICU stay, length of hospital stay, and mechanical ventilation days [31]. There may be at least three reasons for those with better muscle strength to have shorter length of hospitalization. First, patients with better muscle strength can expectorate sputum easier, and they may be discharged earlier if no pneumonia develops secondary to sputum impaction. Second, patients can take their meals in an upright posture if they have sufficient muscle strength, and the upright posture can significantly reduce the risk of choking. Third, patients can get out of bed earlier if they have better strength. The less time patients spend in bed, the fewer complications develop in relation to being bed-ridden. As for ICUMS, the articles exploring its utility in predicting length of hospital stay in COVID-19 patients are still lacking. However, previous studies have mentioned its utility in predicting 90-day survival and discharge destination for ICU patients [21,32]. Our study extrapolated this scale to COVID-19 hospitalized patients, introducing a distinct measure for clinical outcomes prediction. The underlying reason for those with better ICUMS to be discharged earlier may be similar as mentioned above in that patients with better mobility can have fewer complications secondary to being bed-ridden. Further studies are still necessary to confirm its usefulness in COVID-19 patients.

Apart from ICUMS and MRCSS, similar to previous studies, our study also includes patients’ hematology tests to determine if there are certain laboratory data associated with the clinical outcomes we set up. Liu et al. discovered that lymphopenia is predictive of longer hospitalization days [15]. Thiruvengadam et al. observed that a longer length of hospital stay is related to higher LDH, D-dimer, ferritin, and neutrophil–lymphocyte ratio [33]. Moreover, the study performed by Bode et al. revealed that either diabetes or uncontrolled hyperglycemia is a risk factor for longer hospitalization [34]. Nevertheless, none of the relationships between the blood profiles and the established clinical outcomes were found in our study. This may be the result of the relatively small sample sizes in our study. Additionally, we did not stratify our patients according to their disease severity, which may have diminished the relationship between certain laboratory data and the clinical outcomes. On the other hand, to the best of our knowledge, there has been no previous study investigating the association between patients’ hematology tests and their activities of daily living.

Our study has the following limitations. First, the case numbers in this study are relatively small, which in turn limits the number of predictive factors that can be included in the regression model. Second, the cases recruited for our study were from a single hospital, which may narrow the generalizability of our results. Third, patient Barthel index scores three months after discharge were assessed via phone call. Measurement bias on the evaluation of the Barthel index score is thus inevitable.

In conclusion, this study developed a different method to measure and predict COVID-19 patients’ activities of daily living as well as their length of hospital stay. We found that both higher ICUMS and MRCSS are associated with a better Barthel index score three months after discharge and a shorter length of hospitalization as well. Future studies involving more cases that focus on the relationship between COVID-19 patients’ mobility, muscle condition, and clinical outcomes remain necessary.

## Figures and Tables

**Figure 1 healthcare-10-01589-f001:**
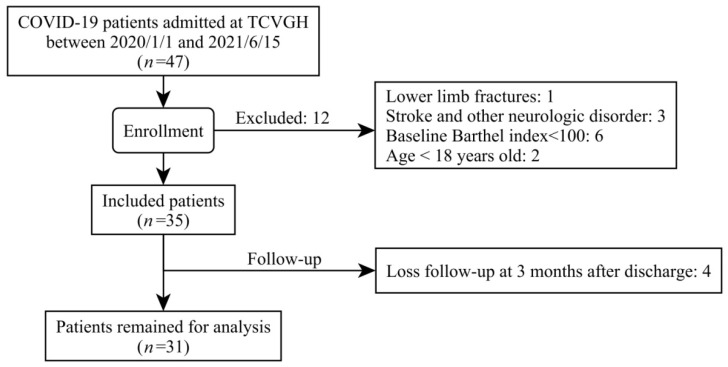
The flowchart for enrollment of our study subjects.

**Figure 2 healthcare-10-01589-f002:**
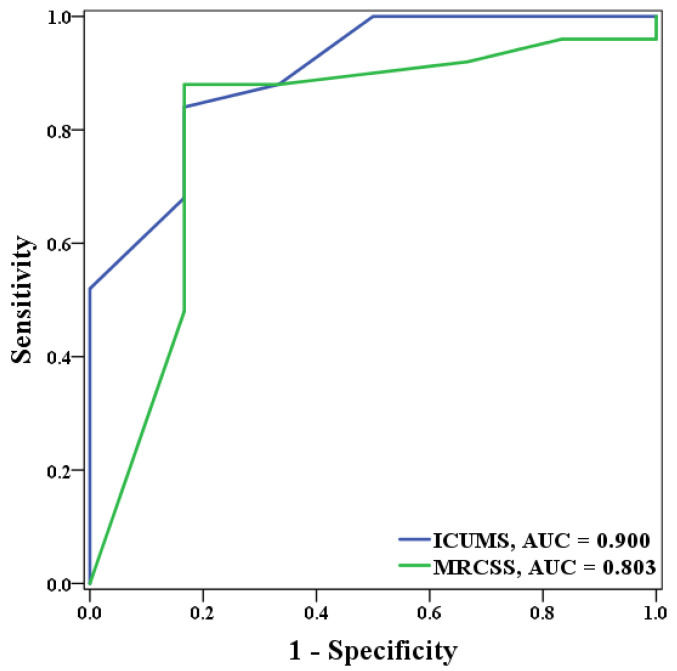
The ROC curve of ICUMS and MRCSS in predicting those with a Barthel index score of less than 100 at three months after discharge.

**Figure 3 healthcare-10-01589-f003:**
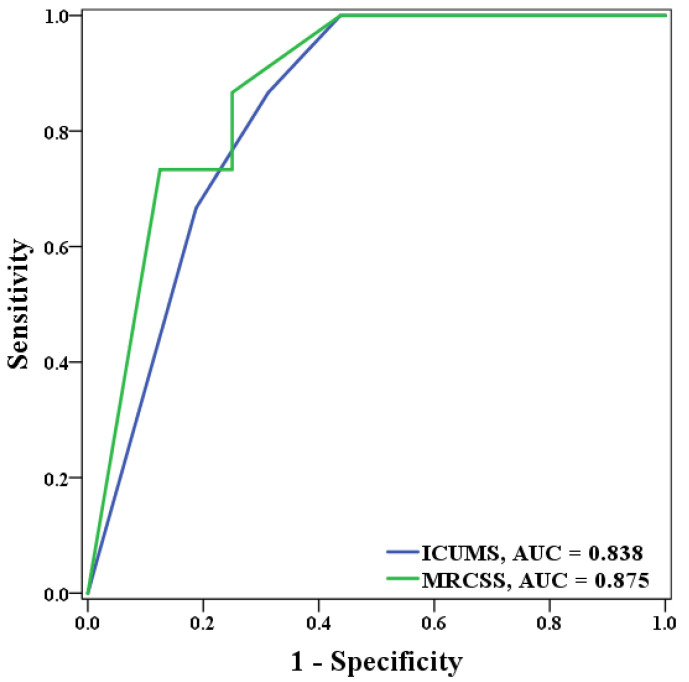
The ROC curve of ICUMS and MRCSS in predicting those hospitalized more than 19 days.

**Figure 4 healthcare-10-01589-f004:**
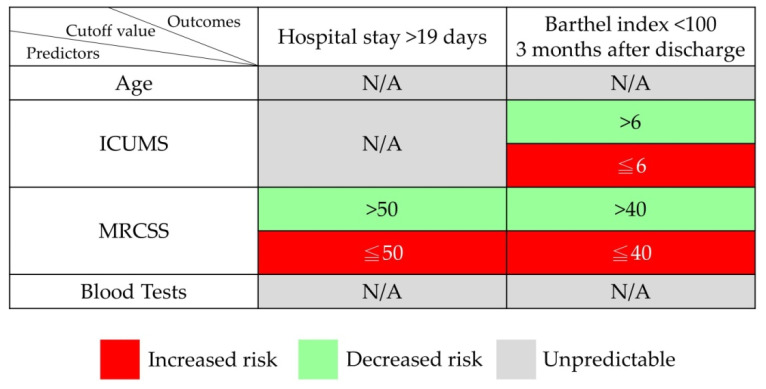
The predictive ability and the cutoff points for patients’ length of hospitalization > 19 days and Barthel index < 100 three months after discharge.

**Table 1 healthcare-10-01589-t001:** The clinical characteristics and the initial profiles of the study subjects.

	Mean	SD	(Mini, Max)	Median
Age (*n* = 31)	51.03	19.0	(18, 84)	51
Barthel index (*n* = 31)	92.9	19.74	(15, 100)	100
Hospitalization (*n* = 31)	23.55	12.07	(8, 56)	20
ICUMS (*n* = 31)	7.23	3.56	(1, 10)	9
MRCSS (*n* = 31)	46.77	16.51	(10, 60)	52
WBC (*n* = 31)	7764.19	4738.53	(2690, 25,410)	7020
Hb (*n* = 31)	13.53	1.41	(10.9, 16.8)	13.4
Total bilirubin (*n* = 29)	0.53	0.28	(0.10, 1.40)	0.50
LDH (*n* = 29)	375	154	(152, 703)	340
D-dimer (*n* = 29)	4.3	9.8	(0.2, 35.2)	0.6
BUN (*n* = 31)	15.4	7.8	(6.0, 44.0)	14.0
Cr (*n* = 31)	0.83	0.21	(0.53, 1.42)	0.77
Na (*n* = 31)	138	5.0	(127, 148)	139
K (*n* = 31)	3.8	0.6	(2.4, 4.9)	3.8
Ca (*n* = 27)	7.5	1.8	(4.0, 9.2)	8.0
P (*n* = 8)	2.9	0.8	(1.6, 4.1)	2.9
Mg (*n* = 11)	2.2	0.2	(1.7, 2.6)	2.2
ESR (*n* = 21)	38.4	28.9	(1.0, 112)	40.0
CRP (*n* = 31)	4.79	6.04	(0.04, 24.3)	2.79
Prolactin (*n* = 30)	0.13	0.14	(0.04, 0.58)	0.07
ALKP (*n* = 29)	70.2	25.5	(20.0, 138)	70.0
APTT (*n* = 28)	32.1	4.1	(23.8, 38.2)	32.4
Prothrombin (*n* = 28)	10.8	0.7	(9.7, 12.4)	10.7
Lactate (*n* = 26)	12.6	6.4	(4.6, 27.1)	10.0
Albumin (*n* = 25)	3.6	0.7	(2.0, 5.1)	3.5
CK (*n* = 30)	145	118	(33.0, 507)	110
CKMB (*n* = 25)	8.1	3.5	(3.0, 15.0)	7.0

ALKP, alkaline phosphatase; APTT, activated partial thromboplastin time; BUN, blood urea nitrogen; Ca, calcium; C, creatine kinase; CKMB, creatine kinase-MB; Cr, creatinine; CRP, C-reactive protein; ESR, erythrocyte sedimentation rate; Hb, hemoglobin; ICUMS, Intensive Care Unit Mobility Scale; K, potassium; LDH, lactate dehydrogenase; Max, maximum; Mg, magnesium; Mini, minimum; MRCSS, Medical Research Council Sum Score; Na, sodium; P, phosphorus; SD, standard deviation; WBC, white blood cell count.

**Table 2 healthcare-10-01589-t002:** The comparison of clinical characteristics and initial blood profiles between the groups based on Barthel index recorded three months after discharge.

	Barthel Index < 100 (Mean ± SD)		*p*-Value
	No (*n* = 25)	Yes (*n* = 6)	
Age	48 ± 18.5	63.8 ± 17.0	0.065
Sex			0.172
Male	11 (44.0%)	5 (83.3%)	
Female	14 (56.0%)	1 (16.7%)	
ICUMS	8.3 ± 2.8	2.8 ± 3.1	0.001 **
MRCSS	51.0 ± 13.7	29.3 ± 16.9	0.015 *
WBC (*n* = 31)	8046 ± 4888	6588 ± 4239	0.369
Hb (*n* = 29)	13.5 ± 1.5	13.5 ± 1.0	0.952
Total bilirubin (*n* = 29)	0.56 ± 0.28	0.37 ± 0.20	0.194
LDH (*n* = 29)	364 ± 147	431 ± 193	0.443
D-dimer (*n* = 29)	4.0 ± 9.6	5.2 ± 11.7	0.238
BUN (*n* = 31)	15.6 ± 8.4	14.5 ± 4.4	0.970
Cr (*n* = 31)	0.84 ± 0.23	0.78 ± 0.15	0.971
Na (*n* = 31)	138.2 ± 5.2	137.3 ± 3.9	0.599
K (*n* = 31)	3.8 ± 0.6	4.0 ± 0.3	0.616
Ca (*n* = 27)	7.3 ± 1.9	8.1 ± 0.5	0.705
P (*n* = 8)	2.8 ± 0.8	3.1 ± 0.9	1.000
Mg (*n* = 11)	2.2 ± 0.3	2.4 ± 0.1	0.127
ESR (*n* = 21)	40.1 ± 32.0	33.0 ± 16.8	0.842
CRP (*n* = 31)	4.47 ± 6.21	6.16 ± 5.58	0.247
Prolactin (*n* = 30)	0.13 ± 0.14	0.16 ± 0.16	0.342
ALKP (*n* = 29)	69.8 ± 24.6	72.4 ± 32.3	0.603
APTT (*n* = 28)	32.3 ± 3.9	31.3 ± 5.5	0.869
Prothrombin (*n* = 28)	10.8 ± 0.7	10.6 ± 0.8	0.610
Lactate (*n* = 26)	12.9 ± 6.6	11.3 ± 6.1	0.670
Albumin (*n* = 25)	3.7 ± 0.7	3.5 ± 0.3	0.830
CK (*n* = 30)	138.6 ± 110.3	173.3 ± 154.2	0.990
CKMB (*n* = 25)	7.9 ± 3.5	9.7 ± 4.5	0.546

* *p* < 0.05; ** *p* < 0.01. ALKP, alkaline phosphatase; APTT, activated partial thromboplastin time; BUN, blood urea nitrogen; Ca, calcium; CK, creatine kinase; CKMB, creatine kinase-MB; Cr, creatinine; CRP, C-reactive protein; ESR, erythrocyte sedimentation rate; Hb, hemoglobin; ICUMS, Intensive Care Unit Mobility Scale; K, potassium; LDH, lactate dehydrogenase; Mg, magnesium; MRCSS, Medical Research Council Sum Score; Na, sodium; P, phosphorus; SD, standard deviation; WBC, white blood cell count.

**Table 3 healthcare-10-01589-t003:** The comparison of clinical characteristics and initial blood profiles between the groups based on the length of hospital stay.

	Hospitalization > 19 Days (Mean ± SD)		*p*-Value
	No (*n* = 15)	Yes (*n* = 16)	
Age	44.1 ± 20.7	57.5 ± 15.2	0.044 *
Sex			0.862
Male	7 (46.7%)	9 (56.3%)	
Female	8 (53.3%)	7 (43.8%)	
ICUMS	9.5 ± 0.7	5.1 ± 3.8	0.001 **
MRCSS	57.3 ± 4.7	36.9 ± 17.6	<0.001 **
WBC (*n* = 31)	7701 ± 5838	7823 ± 3618	0.513
Hb (*n* = 31)	13.9 ± 1.4	13.2 ± 1.3	0.198
Total bilirubin (*n* = 29)	0.60 ± 0.32	0.46 ± 0.22	0.236
LDH (*n* = 29)	337 ± 132	412 ± 168	0.285
D-dimer (*n* = 29)	1.3 ± 1.8	7.1 ± 13.2	0.739
BUN (*n* = 31)	14.0 ± 7.1	16.6 ± 8.4	0.190
Cr (*n* = 31)	0.79 ± 0.17	0.86 ± 0.25	0.661
Na (*n* = 31)	137.9 ± 3.7	138.1 ± 5.9	0.674
K (*n* = 31)	3.8 ± 0.5	3.9 ± 0.3	0.427
Ca (*n* = 27)	7.7 ± 1.7	7.2 ± 1.8	0.272
P (*n* = 8)	2.5 ± 0.9	3.2 ± 0.7	0.464
Mg (*n* = 11)	2.0 ± 0.3	2.3 ± 0.2	0.285
ESR (*n* = 21)	41.8 ± 27.2	35.8 ± 31.0	0.422
CRP (*n* = 31)	4.0 ± 6.2	5.5 ± 6.0	0.264
Prolactin (*n* = 30)	0.09 ± 0.07	0.18 ± 0.18	0.069
ALKP (*n* = 29)	66.9 ± 18.5	73.3 ± 30.9	0.821
APTT (*n* = 28)	32.3 ± 4.3	31.8 ± 4.1	0.744
Prothrombin (*n* = 28)	10.7 ± 0.6	10.9 ± 0.8	0.579
Lactate (*n* = 26)	13.0 ± 7.3	12.1 ± 5.4	0.752
Albumin (*n* = 25)	3.9 ± 0.6	3.4 ± 0.6	0.088
CK (*n* = 30)	118.6 ± 72.9	172.5 ± 148.5	0.690
CKMB (*n* = 25)	7.9 ± 4.0	8.3 ± 2.9	0.651

* *p* < 0.05; ** *p* < 0.01. ALKP, alkaline phosphatase; APTT, activated partial thromboplastin time; BUN, blood urea nitrogen; Ca, calcium; CK, creatine kinase; CKMB, creatine kinase-MB; Cr, creatinine; CRP, C-reactive protein; ESR, erythrocyte sedimentation rate; Hb, hemoglobin; ICUMS, Intensive Care Unit Mobility Scale; K, potassium; LDH, lactate dehydrogenase; Mg, magnesium; MRCSS, Medical Research Council Sum Score; Na, sodium; P, phosphorus; SD, standard deviation; WBC, white blood cell count.

**Table 4 healthcare-10-01589-t004:** The odds ratio of a Barthel index scored less than 100 based on age, ICUMS and MRCSS.

	Univariate			Adjusted for Age		
	OR	95%CI	*p*-Value	OR	95%CI	*p*-Value
Age	1.05	(0.99–1.12)	0.081			
ICUMS	0.62	(0.44–0.88)	0.007 **	0.61	(0.41–0.90)	0.012 *
MRCSS	0.92	(0.87–0.98)	0.012 *	0.93	(0.87–0.99)	0.024 *

* *p* < 0.05; ** *p* < 0.01; CI, confidence interval; ICUMS, Intensive Care Unit Mobility Scale; MRCSS, medical council research sum score; OR, odds ratio.

**Table 5 healthcare-10-01589-t005:** The odds ratio of a hospital stay > 19 days based on age, ICUMS, and MRCSS.

Univariate				Adjusted for Age		
	OR	95%CI	*p*-Value	OR	95%CI	*p*-Value
Age	1.04	(1.00–1.09)	0.058			
ICUMS	0.46	(0.21–1.02)	0.055			
MRCSS	0.84	(0.72–0.97)	0.017 *	0.82	(0.69–0.97)	0.02 *

* *p* < 0.05; CI, confidence interval; ICUMS, Intensive Care Unit Mobility Scale; MRCSS, Medical Council Research Sum Score; OR, odds ratio.

**Table 6 healthcare-10-01589-t006:** The AUC, cutoff points, sensitivity, specificity, PPV, NPV, and accuracy of ICUMS and MRCSS to predict patients with Barthel index < 100 at three months after discharge and hospitalization > 19 days.

		AUC	Cut Point	Sensitivity	Specificity	PPV	NPV	Accuracy
Barthel index < 100	ICUMS	0.900	6	84.0%	83.3%	95.5%	55.6%	83.9%
	MRCSS	0.803	40	88.0%	83.3%	95.7%	62.5%	87.1%
Hospitalization > 19 days	ICUMS	0.838	6	100%	56.3%	68.2%	100%	77.4%
	MRCSS	0.875	50	86.7%	75.0%	76.5%	85.7%	80.6%

AUC, area under curve; ICUMS, Intensive Care Unit Mobility Scale; MRCSS, Medical Council Research Sum Score; PPV, positive predictive value; NPV, negative predictive value.

## Data Availability

The datasets generated and analyzed during the current study are available through the corresponding author upon reasonable request.

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
