# Peer review of "Predictors of Basic Activity in Daily Living and Length of Hospitalization in Patients with COVID-19"

_healthcare, 2022, doi:10.3390/healthcare10081589_

Round 1

Reviewer 1 Report

The manuscript "Predictors of Basic Activity in Daily Living and Length of Hospitalization in Patients with COVID-19" has evaluated the association of physical activity with length of hospital stay. The conclusion is supported by the data provided in the manuscript. The data provided has evaluated and correlated the length of hospital stay with physical activity and blood parameters. However, as the title states the evaluation of "predictors" of basic physical activity has not been done. The study has correlated physical activity with hospital stay, so please change the title or analyze/correlate the blood parameters with physical activity, which has been discussed only briefly in one paragraph  at the end of discussion.

Author Response

Thanks for the reviewer’s affirmative comments. About the blood parameters in our study, as shown in Table 2 and Table 3, there are no significant differences in all the blood profiles between those with Barthel index scored 100 three monthes after discharge and those without, and between those hospitalized for more than 19 days and those hospitalized less than 19 days. Therefore, these blood parameters were not included in further logistic regression analysis. According to the reviewer’s suggestion, we tried to analyze the correlation between the blood parameters with physical activity in our study, which is shown in Supplemental Table 4. As expected, no significant correlation could be identified between any of them. Therefore, according to the results of our study, we can only “predict” the basic activity in daily living (evaluated by Barthel index) and length of hospitalization by physical activity status (ICUMS) and muscle strength (MRCSS) instead of blood parameters.

Reviewer 2 Report

The topic of the manuscript is within the scope of the Journal and could be relatively valuable to the scientific audience. The quality of the research design is acceptable.

TITLE

The title of the article is accurate.

ABSTRACT

Abstract reflects the work done and the conclusions drawn.

INTRODUCTION

The introduction is well prepared but authors should provide justification for the research hypothesis (What specific sources are used to formulate the hypothesis?).

METHOD

Some clarifications are however needed. Please address the following issues:

Have the assumptions for logistic regression been tested and met?

For instance, absence of multicollinearity, and lack of strongly influential outliers?

RESULTS

Correlations between variables, which were recruited for logistic regression, could be shown in a table.

DISCUSSION

In the discussion section authors make sufficient focus on comparing your findings to results of previous studies. Conclusions are justified by the experimental evidence.

TO SUM UP I think the author(s) need to make the recommended corrections.

Author Response

Thanks for the reviewer’s detailed examination of our paper and the valuable comments, which help us to present our study more clearly. The manuscript has been revised based on these suggestions. Our responses to the reviewer’s comments are given below on a point-by-point basis.

  1. We agree that we should provide the justification for the research hypothesis, and the last paragraph of the introduction was revised as following: “Based on a past study,[1] which showed impaired baseline activities of daily living being associated with functional decline at discharge, we hypothesize that COVID-19 patients with a poor physical activity performance upon admission shall have longer hospital stays and worse functional outcomes.”
  2. Thanks for the reviewer’s reminding about the prerequisite factors of logistic regression. In our study, we used two clinical measurement tools, MRCSS and ICUMS, to evaluate patients’ muscle strength and physical activity status, respectively. Because of muscle strength of four limbs greatly influence the performance of physical activity, it is inevitable that MRCSS was highly associated with ICUMS in our study. As shown in supplemental Table 3, the correlation coefficient between MRCSS and ICUMS is high. Therefore, we did not use them to statistically adjust each other in our regression model. Only “age” was used in the statistical adjustment. About the strongly influential outliers, we provided supplemental Table 1 and 2 to show the range of data of our participants. No significant outliers could be identified in our study cases.
  3. Thanks for the reviewer’s valuable suggestion. We performed Spearman's rho test to examine the correlation between each of the two variables among age, MRCSS, ICUMS, length of hospitalization, and Barthel index score three months after discharge, and the data was shown in Supplemental Table 3.
  4. Thanks for the reviewer’s detailed examination of our manuscript and affirmative comments.

Reference:

  1. Hosoda, T.; Hamada, S., Functional decline in hospitalized older patients with coronavirus disease 2019: a retrospective cohort study. BMC geriatrics 2021, 21 (1), 1-9.

Reviewer 3 Report

Dear authors,

Thanks for the submission. Even though the idea seems to be promising, the way it is presented is very hard for me to follow. 

Intro is a bit confusing and hard to follow to get a good idea about the problem and purpose of the study. 

Methods are seemingly okay, but still the use of these scales in COVID19 patients is questionable. Validity and reliability issues are present affecting the results and discussion. 

Stats look okay, assuming that these are the IV and DVs and they match the RQ. 

Results are difficult to follow, and the tables are very confusing. 

Discussion is somehow okay, assuming that Stats and Results are appropriate, but I didnt see any "real discussion" on the results. Just we found this, others found that so we all are on the same page, no mechanistic or teleological explanation. 

Author Response

Thanks for the reviewer’s detailed examination of our paper and the valuable comments, which help us to present our study more clearly. The manuscript has been revised based on these suggestions. Our responses to the reviewer’s comments are given below on a point-by-point basis.

  1. We agree that our introduction section was not clear enough regarding to the current problem and purpose of our study. Therefore, we revised our paragraph as following: “Physical activity can be evaluated through several simple and easy clinical measurement tools by healthcare workers with no additional cost. However, according to our literature review as above, there were still few studies regarding the predictive ability of physical activity on the functional prognosis and length of hospitalization until now. Therefore, in this retrospective cohort study, we aim to find out whether a COVID-19 patient’s physical activity upon admission is associated with both functional prognosis and hospitalization stay duration. Additionally, we want to identify the value of activity status, along with other patient characteristics and laboratory data, in predicting clinical outcomes of COVID-19 patients. We hypothesize that COVID-19 patients with a poor physical activity performance upon admission shall have longer hospital stays and worse functional outcomes.”
  2. Indeed, there have been no studies in the past to use ICUMS and MRCSS on patients with COVID-19. However, both ICUMS and MRCSS were validated to evaluate the physical performance and the degree of muscle weakness in ICU patients.[1,2] Since all patients recruited in this study had pneumonia appearance on chest X-rays and prominent clinical symptoms related to COVID-19, we thus designed this study to use ICUMS and MRCSS in this group of patients with novelty. Considering the rapid progressing nature of the earlier strain of SARS-CoV2 in the patients we recruited, we believe these two measurements could be used in these patients with COVID-19 pneumonia.
  3. Thanks for the reviewer’s comment on our statistical analysis. As shown in the supplemental table 3, the correlations between the independent variables and dependent variables are significant. Therefore, the odds ratio in the logistic regression can be high, as expected. Furthermore, high correlations could be observed between our independent variables, ICUMS and MRCSS. Consequently, they were not used to statistically adjust each other in our regression model. Only “age” was used to statistically adjust the odds ratio.
  4. In order to make our results clearer to readers, we added supplemental tables 1-3 to better reflect the characteristics of our participants. Furthermore, we added some sentences to describe the results better as following: “The results of differences between the two outcomes were similar. Both ICUMS and MRCSS were significantly different between those with longer and shorter hospitalization, and between those with full Barthel index scores three months after discharge and those without. While age was significantly different only between those with longer and shorter hospitalization, all the differences of sex and the blood profiles were not significant in both outcomes in our study.” Regarding to the results of logistic regression analysis and ROC curve analysis, we added Figure 4 as a summary to present our predictive model more clearly.
  5. We acknowledge that our previous manuscript was insufficient in the discussion of underlying mechanism. Therefore, the sentences were added as following in order to better address this concern: “The mechanism behind may be related to the overwhelming inflammatory responses caused by severe COVID-19. The more declined physical activity in the initial stage of COVID-19 signified the more advanced disease severity, which could therefore bring subsequent longer length of hospitalization and more physical sequelae three months after discharge.” Thanks again for your critical comments.

Reference:

  1. Hodgson, C.; Needham, D.; Haines, K.; Bailey, M.; Ward, A.; Harrold, M.; Young, P.; Zanni, J.; Buhr, H.; Higgins, A., Feasibility and inter-rater reliability of the ICU Mobility Scale. Heart & Lung 2014, 43 (1), 19-24.
  2. Hermans, G.; Clerckx, B.; Vanhullebusch, T.; Segers, J.; Vanpee, G.; Robbeets, C.; Casaer, M. P.; Wouters, P.; Gosselink, R.; Van Den Berghe, G., Interobserver agreement of Medical Research Council sum‐score and handgrip strength in the intensive care unit. Muscle & nerve 2012, 45 (1), 18-25.

Reviewer 4 Report

Dear authors, 

The study is exceptional.

I congratulate you for venturing to create predictive models that allow us to improve intervention times and lines of work.

Given the results found, I encourage you to visually create a predictive model that allows us to detect which set of variables are most appropriate to include in this model.

I suggest incorporating this joint and synthesis figure at the end of the results section. 

Kind regards, 

Author Response

After thorough discussion, we agree that we should create a summarized figure to present our predictive model more clearly. We have added it at the end of the results section. Thank you very much for the critical suggestion!

Reviewer 5 Report

Dear authors,

Manuscript is titled:  Predictors of Basic Activity in Daily Living and Length of Hospitalization in Patients with COVID-19

ICUMS and MRCSS are the main point of the manuscript to evaluate movement and muscle strength. Both must be explained at methodology section.

Why patients with Barthel index score less than 100 are excluded? Explained your reasons please.

Age ranged from 6-84 years and sample size of 33, both aspects are possible bias that limit the external validity of the sample. Children has not the same comorbidities or associated pathologies that a person with 80 years. Data should be analyzed by age too.

Although differences between groups have been explained, this data could influence the results again. Could you explain these differences better?

Although the topic of the manuscript is predictors of basic activity daily living, few references mention these activities. More literature should support this aspect and also, you should establish better the differences between the activities during hospitalization and the activities after discharge.

Author Response

Thanks for the reviewer’s detailed examination of our paper and the valuable comments, which help us to present our study more clearly. The manuscript has been revised based on these suggestions. Our responses to the reviewer’s comments are given below on a point-by-point basis.

  1. Indeed, ICUMS and MRCSS are the main points and focused predictors of the manuscript. We have explained both in the paragraph of “2.2 study design” of our materials and methods as following: “The ICUMS was first developed by Hodgson et al. to evaluate the highest level of mobility in adult ICU (Intensive Care Unit) patients. The scale is ranked from zero to ten, where zero means no active movement at all by the patient, and ten means the patient can walk independently without any aid. Tipping et al. observed the predictive value of the ICUMS for ICU discharge survival and discharge destination. The MRCSS was initially used to assess muscle weakness in patients with Guillan-Barre syndrome. Hermans et al. also found its reliability and reproducibility in evaluating muscle weakness in ICU patients. It was also used to predict discharge destination in the study done by Perme et al. The score is composed of six muscle strengths on each side, with each muscle being scored from zero to five, resulting a total score from zero to sixty.”
  2. Those with a baseline Barthel index score less than 100 before the onset of COVID-19 were excluded from our study, because one of our outcome measurements was Barthel index scores three months after discharge. If the baseline scores of the participants were different, it would become a significant confounding factor of Barthel index scores three months after discharge.
  3. We acknowledge that children may have a completely different pathophysiology from adults. Therefore, our data were re-analyzed after excluding two cases who were below 18 years old. Furthermore, we provide our data stratified by age groups 18-40, 41-63, and 64-84 in supplemental Table 1 & 2. Thanks for your critical comment.
  4. We tried to explain these differences between the groups in the result section as following: “The results of differences between the two outcomes were similar. Both ICUMS and MRCSS were significantly different between those with longer and shorter hospitalization, and between those with full Barthel index scores three months after discharge and those without. While age was significantly different only between those with longer and shorter hospitalization, all the differences of sex and the blood profiles were not significant in both outcomes in our study.”
  5. We agree that the activities of daily living, which is assessed by Barthel Index in our manuscript, should be emphasized and discussed more in our study. Therefore, the following paragraphs were added in the section 2 of discussion: “One of the most important functional outcomes, the basic activity in daily living, was assessed by Barthel index in our study. Some items of Barthel index were important for hospitalized patients, such as feeding, transfer, toileting, bowel management and bladder control. Other items were more essential for discharged patients without caregivers, including grooming, dressing, mobility, and stairs climbing. Barthel index has been used as an important indicator of hospital outcome,[1] and could be used as a predictor of mortality.[2] On the other hand…”

References:

  1. Ocagli H, Cella N, Stivanello L, Degan M, Canova C. The Barthel index as an indicator of hospital outcomes: A retrospective cross-sectional study with healthcare data from older people. Journal of Advanced Nursing. 2021 77(4):1751-1761.
  2. Katano S, Yano T, Ohori K, Kouzu H, Nagaoka R, Honma S, Shimomura K, Inoue T, Takamura Y, Ishigo T, Watanabe A, Koyama M, Nagano N, Fujito T, Nishikawa R, Ohwada W, Hashimoto A, Katayose M, Ishiai S, Miura T. Barthel Index Score Predicts Mortality in Elderly Heart Failure - A Goal of Comprehensive Cardiac Rehabilitation. Circulation Journal. 2021 24;86(1):70-78.

Round 2

Reviewer 3 Report

Dear authors,

Thanks for the resubmission and the clarifications. I still think that the article is promising, but the writing is problematic.

The intro still confuses me, as the whole idea is around basic physical functioning, but the Intro goes around the physical activity (PA). PA is not the same as basic functioning or mobility etc, but these terms are used interchangeably and this is incorrect. Intro needs to connect the concepts with the tools used in the Methods something that still I dont see it. Tools need to described in more detail and clarifications why in some cases means or medians are used and what is the rationale for the cutoff points.

Results are probably okay, but since the hematological provide no extra info do we need them? Also i dont follow how in the discussion the explanation about the inflammatory status due to COVID is presented while none of the related inflammatory markers was different between the groups.

Author Response

Thanks for the reviewer's critical comments to our manuscript. We had made significant changes throughout our paragraph to address the issues concerned by the reviewer, and we believe it can definitely make our study better and easier to understand by the readers of Healthcare.

  1. We acknowledge that the word "physical activity" should not be used interchangeably with "physical functioning". Actually we tried to predict the patients' functional outcome by their physical activity status during initial hospitalization. In order to avoid the misunderstanding, we replaced the word "physical activity" to "physical performance" throughout the paragraph. Thanks for your crucial comment.
  2. We agree that the introduction should connect the tools mentioned in the method, and the paragraph of introduction was revised as following: "... van Gassel et al. observed that COVID-19 patients with better muscle strength and pulmonary function experience better physical functioning three months after discharge. The outcome measurement used to assess muscle strength in this study was Medical Research Council Sum Score (MRCSS), which is the sum score of both proximal and distal muscle strength of four limbs. This score can provide a comprehensive strength evaluation of extremities, and could be one of the possible predictors of physical function in the future. On the other hand, the study done by Pinto et al. drew a contradicting conclusion, determining that physical activity, which was evaluated by Baecke Questionnaire of Habitual Physical Activity (BQHPA), has no independent association with clinical outcomes among hospitalized COVID-19 patients. In contrast to BQHPA, which evaluates individual's habitual physical activities over the previous 12 months, Intensive Care Unit Mobility Scale (ICUMS) reflects the physical mobility at the timing of hospital admission, and could be another predictive factor of functional outcomes. Because of the rarity of literature in the past, the ability of physical performance in predicting functional outcomes in COVID-19 patients deserves more studies to clarify."
  3. We agree that the clinical tools in our study should be described in more detail as following: "…ICUMS…The scale is ranked from zero to ten, where zero means no active movement at all by the patient, and ten means the patient can walk independently without any aid. The criteria of scoring one to nine were capable of performing the following works: 1. Bed activities. 2. Passive moving to a chair. 3. Sitting over edge of bed. 4. Standing. 5. Active transferring to a chair. 6. Marching on spot. 7. Walking with assistance of two or more people. 8. Walking with assistance of one person. 9. Independent walking with a gait aid. … . The MRCSS, on the other hand, is composed of the muscle strength of moving the bilateral shoulders, elbows, wrists, hips, knees, and ankles, with each muscle being scored from zero to five, resulting a total score from zero to sixty."
  4. In Table 1, the mean and median values were both provided in order to present the data distribution of the enrolled patients in the study. In Table 2 and 3, only the mean value and standard deviation were shown in order to reduce the complexity of the tables.
  5. We agree that the rationale for the cutoff points should be described in more detail, and we revised the paragraph as following: "...which means the COVID-19 patients might be hospitalized for more than 19 days or has a Barthel index < 100 three months after discharge if their initial ICUMS score was less than six. On the other hand, for MRCSS, it was 40 and 50, respectively. It indicates that patients might not be totally independent three months after discharge if their initial MRCSS scored less than 40, and might be hospitalized for more than 19 days if their initial MRCSS scored less than 50."
  6. We believe that the hematological results are still needed in our study, since some evidences in the past have indicated that some blood profiles such as LDH, D-dimer, ferritin, blood sugar and neutrophil-lymphocyte ratio can be predictive of length of hospitalization. [1,2] Our results were different from the past, and we think they should be disclosed in this manuscript.
  7. We're sorry that the proposed inflammatory mechanism should not be present in our manuscript since our data did not support it. Therefore, we revised the paragraph as following: "The mechanism behind may be related to the detrimental involvement of multiple organs in addition to the respiratory system caused by severe COVID-19. The more declined physical performance in the initial stage of COVID-19 signified the more advanced disease severity, which could therefore bring subsequent longer length of hospitalization and more physical sequelae three months after discharge."

References:

  1. Thiruvengadam, G.; Lakshmi, M.; Ramanujam, R., A study of factors affecting the length of hospital stay of COVID-19 pa-tients by Cox-proportional hazard model in a South Indian tertiary care hospital. Journal of Primary Care & Community Health 2021, 12, 21501327211000231.
  2. Bode, B.; Garrett, V.;  Messler, J.;  McFarland, R.;  Crowe, J.;  Booth, R.; Klonoff, D. C., Glycemic characteristics and clinical outcomes of COVID-19 patients hospitalized in the United States. Journal of diabetes science and technology 2020, 14 (4), 813-821.

Reviewer 5 Report

Dear authors, 

Thank you for considering my suggestions, the manuscript is better now.

Author Response

We're very grateful for the reviewer's previous comments and suggestions. Thank you!

Round 3

Reviewer 3 Report

Dear authors, 

Thanks for the revisions. The problem that I see is that the more I read it, the more clarifications I ask, the more problems I recognize that make the whole effort problematic. 

This is the 3rd time that I am struggling to understand the concepts, the design, the methodology etc. For example you say that you have 2 groups, early or late hospitalization and BI more or less 100. These are 4 groups and you present results between 2 not 4. Also never explained again why you selected these criteria to as cuttoff points. 

Author Response

Thanks for the reviewer’s detailed examination on our manuscript. After significant revisions in the last version of our manuscript, we’ve addressed 7 issues the reviewer concerned the most. We’re very grateful for the reviewer’s critical suggestions that have made our concepts, design, and methodology clearer than before. However, we’re sorry that there’re still 2 issues needs to clarify, and they’re listed below:

  • Q1: You say that you have 2 groups, early or late hospitalization and BI more or less 100. These are 4 groups and you present results between 2 not 4.
    Answer:
    In the last paragraph of section 2.2 “study design”, we clearly described that 2 targets were selected as the outcomes of our study, which were “the length of hospital stay” and “the Barthel index scores three months after discharge”. We then compared the data between those hospitalized for more than 19 days and those hospitalized less than 19 days, and also, between those with a Barthel index score of 100 and less than 100 three months after discharge, which were shown in Table 3 and Table 2, respectively. It is taken for granted to compare the data between the 2 groups according to the outcomes we selected, instead of 4 altogether. After all, it is meaningless to compare the data between the irrelevant outcomes, for example, data between patients with short hospitalization and patients scored 100 at Barthel index.
  • Q2: Explain again why you selected these criteria to as cutoff points.
    Answer:
    The reason for the cutoff points selected to divide the patients into 2 groups was clearly described In the last paragraph of section 2.2 as following: “Patients were divided into two groups according to their outcomes, those with a hospitalization period longer than median or shorter than median, and those with a Barthel index score of 100 (which means completely independent) or less than 100 (which means partially dependent) three months after discharge.” On the other hand, the rationale for the cutoff points of our results was described in more detail in the last version of revision as following: "...which means the COVID-19 patients might be hospitalized for more than 19 days or has a Barthel index < 100 three months after discharge if their initial ICUMS score was less than six.” and “It indicates that patients might not be totally independent three months after discharge if their initial MRCSS scored less than 40, and might be hospitalized for more than 19 days if their initial MRCSS scored less than 50." Furthermore, we’ve added Figure 4 to better delineate the design and results of our study. Please check the efforts we made on our revisions, and we believe they can make the readers understand the whole study better.